# Antimicrobial and Immunoregulatory Activities of TS40, a Derived Peptide of a TFPI-2 Homologue from Black Rockfish (*Sebastes schlegelii*)

**DOI:** 10.3390/md20060353

**Published:** 2022-05-26

**Authors:** Hongmei Liu, Guanghua Wang, Dongfang Hao, Changbiao Wang, Min Zhang

**Affiliations:** 1School of Marine Science and Engineering, Qingdao Agricultural University, Qingdao 266109, China; 20192211554@stu.qau.edu.cn (H.L.); 200401105@qau.edu.cn (G.W.); 20192111569@stu.qau.edu.cn (D.H.); 20202211008@stu.qau.edu.cn (C.W.); 2Laboratory for Marine Biology and Biotechnology, Pilot National Laboratory for Marine Science and Technology, Qingdao 266109, China

**Keywords:** *Sebastods schegelii*, TFPI-2, antibacterial, antiviral, immunomodulatory

## Abstract

Tissue factor pathway inhibitor-2 (TFPI-2) is a Kunitz-type serine protease inhibitor. Previous reports have shown that TFPI-2 plays an important role in innate immunity, and the C-terminal region of TFPI-2 proved to be active against a broad-spectrum of microorganisms. In this study, the TFPI-2 homologue (*SsTFPI-2*) of black rockfish (*Sebastods schegelii*) was analyzed and characterized, and the biological functions of its C-terminal derived peptide TS40 (FVSRQSCMDVCAKGAKQHTSRGNVRRARRNRKNRITYLQA, corresponding to the amino acid sequence of 187-226) was investigated. The qRT-PCR (quantitative real-time reverse transcription-PCR) analysis showed that the expression of *SsTFPI-2* was higher in the spleen and liver. The expression of *SsTFPI-2* increased significantly under the stimulation of *Listonella anguillarum*. TS40 had a strong bactericidal effect on *L**. anguillarum* and *Staphylococcus aureus*. Further studies found that TS40 can destroy the cell structure and enter the cytoplasm to interact with nucleic acids to exert its antibacterial activity. The in vivo study showed that TS40 treatment could significantly reduce the transmission of *L. anguillarum* and the viral evasion in fish. Finally, TS40 enhanced the respiratory burst ability, reactive oxygen species production and the expression of immune-related genes in macrophages, as well as promoted the proliferation of peripheral blood leukocytes. These results provide new insights into the role of teleost TFPI-2.

## 1. Introduction

In the whole aquaculture industry, the prevention and control of fish diseases is the key factor to determine the success or failure of aquaculture [1]. In the past, the use of antibiotics caused the emergence of drug-resistant strains, drug residues and obstacles to the immune system in aquatic animals and even the imbalance of the microecological environment [2]. Antimicrobial peptides (AMPs), which have been found from organisms to humans [3,4,5], are a new type of antimicrobial agent. Aquatic animals in the natural environment produce AMPs through innate immunity to protect themselves from pathogenic microorganisms [6,7]. AMPs have a broad application prospect because of their low drug resistance rate, wide antibacterial spectrum and unique antibacterial mechanism [8,9].

Two members of the tissue factor pathway inhibitor (TFPI) family [10,11], TFPI-1 and its analog TFPI-2, are Kunitz-serine protease inhibitors that reversibly regulate blood coagulation [12,13,14]. Similar to TFPI-1, TFPI-2 has three Kunitz-type domains (KDs) distributed at the negatively charged N-terminus and the positively charged C-terminus [15]; these three kunitz domains have different functions: the first can bind and inhibit the function of the TF/VI complex, the second can bind and inhibit the function of the Xa factor and the last domain is closely related to the metabolism of TFPI-2 in the body [16,17]. Studies have showed that the truncated peptide EDC34 resulting from the C-terminal truncation of human TFPI-2 has antibacterial activity under physiological buffer conditions [18]. Subsequently, TFPI-2 C-terminal peptides of several vertebrates, including chimpanzees (*Pan troglodytes*), mice (*Mus musculus*), chickens (*Gallus gallus*), frogs (*Xenopus tropicalis*), turtles (*Alligator mississippiensis*) and sharks (*Callorhinchus milii*), were shown to have antibacterial activity against *Escherichia coli* and *Pseudomonas aeruginosa* [19] and have great potential in antimicrobial therapy. However, the study of TFPI-2 in fish is limited to zebrafish (*Danio rerio*) [20], red drum (*Sciaenops ocellatus*) [21], half-smooth tongue sole (*Cynoglossus semilaevis*) [12,13] and Japanese flounder (*Paralichthys olivaceus*) [22].

Black rockfish, *Sebastods schegeli*, is one of the common economic fish species in the Yellow Sea and Bohai Sea of China [23]. In this study, we characterized a TFPI-2 gene (*SsTFPI-2*) from black rockfish and conducted research on its expression pattern; synthesized its C-terminal peptide TS40 and studied its antibacterial effect, antiviral effect and immunomodulatory functions.

## 2. Results

### 2.1. Sequence Characterization of SsTFPI-2

The open reading frame (ORF) of *SsTFPI-2* is 681 bp, encoding 226 amino acids. Its predicted molecular weight was 26.01 kDa, with a theoretical pI of 9.14 (Table 1). It contains a signal peptide sequence (residues 1–20), three KDs (residues 25–78, 85–138 and 145–198, respectively) and a region of a low compositional region (LCR, residues 207–220) (Figure 1 and Figure A1). The secondary structure analysis showed that there are 10 α helices, eight extending chains and 10 random coils (Figure A1). The amino acids of SsTFPI-2 were modeled by Phyre2 using c4bd9B (carboxypeptidase inhibitor smci) as a template sequence; the results showed that there is 73% sequence similarity, and SsTFPI-2 was mainly composed of a random curl and α helix, which was basically consistent with the prediction of secondary structure (Figure A1). The BLAST analysis revealed that SsTFPI-2 shares overall sequence identities with the TFPI-2 of a number of teleost species. Specifically, SsTFPI-2 was 97.78%, 84.07%, 82.74%, 85.40%, 80.97%, 84.51% and 80.53% identical to the TFPI-2 of *Sebastes umbrosus*, *Perca fluviatilis*, *Sciaenops ocellatus*, *Etheostoma cragini*, *Collichthys lucidus*, *Chelmon rostratus* and *Perca flavescens*; the overall sequence identity between SsTFPI-2 and human TFPI-2 is 51.11% (Figure A2). Through the construction of the SsTFPI-2 neighbor joining the phylogenetic tree, it was found that SsTFPI-2 is closest to the TFPI-2 of honeycomb rockfish (*Sebastes umbrosus*) (Figure A3).

### 2.2. Expression of SsTFPI-2 under Normal Physiological Conditions

A qRT-PCR (quantitative real-time reverse transcription-PCR) was carried out to detect the expression profiles of *SsTFPI-2* in various tissues of black rockfish under normal physiological conditions. According to the results, the expression of *S**sTFPI-2* was widespread in all examined tissues. *S**sTFPI-2* expression was detected in muscles, gills, blood, kidney, intestine, brain, spleen and liver, in increasing order (Figure 2).

### 2.3. Expression of SsTFPI-2 under a Bacterial Challenge

In order to examine the expression patterns of SsTFPI-2 after fish pathogen infection, we conducted an experimental challenge on black rockfish with *Listonella anguillarum* and detected the expression of *SsTFPI-2* in the liver, spleen and head kidney. The results showed that *SsTFPI-2* expression was up-regulated in the liver, spleen and kidney after infection, and the peaks were reached at 8 h (16.4 times), 4 h (39.8 times) and 8 h (3.6 times), respectively (Figure 3).

### 2.4. Determination of Peptide Sequence

Predicted by the characteristics including charge number, isoelectric point and hydropathicity, etc. of the amino acid sequence at the C-terminus of SsTFPI-2 (Table 1), TS40 (FVSRQSCMDVCAKGAKQHTSRGNVRRARRNRKNRITYLQA), corresponding to the amino acid sequence of 187–226, was synthesized and further tested for biological functions.

### 2.5. Antibacterial Activity of TS40

As measured by the antibacterial spectrum assay, TS40 has an effect on Gram-positive bacteria *Staphylococcus aureus* and *Streptococcus agalactiae* and Gram-negative bacteria *L. anguillarum* and *Vibrio parahaemolyticus*. To further clarify the antibacterial activity of TS40 against target bacteria, the Minimum inhibitory concentration (MIC) and minimum bactericidal concentration (MBC) were tested. The results indicated that the MICs of TS40 against *S. aureus*, *L. anguillarum*, *V. parahaemolyticus* and *S. agalactiae* were 12.5 μM, 25 μM, 400 μM and 800 μM, respectively (Table 2). The MBCs of TS40 against *S. aureus* and *L. anguillarum* were 25 μM and 50 μM, respectively, and the MBCs of TS40 against *S. agalactiae* and *V. parahaemolyticus* were >800 μM (Table 2). Otherwise, P86P15 had no effect on the viability of bacterial cells.

### 2.6. The Killing Kinetics of TS40

We further investigated the bactericidal kinetics of TS40 against *S. aureus* and *L. anguillarum* (Figure 4). The results showed that the killing rates of TS40 against *S. aureus* and *L. anguillarum* were similar in general. Specifically, after 2 h of treatment by TS40, the killing rates of *S. aureus* and *L. anguillarum* were 87.84% and 70.89%, respectively; after 4 h treatment by TS40, killing effects of 98.7% and 99.9% were achieved for *S. aureus* and *L. anguillarum*, respectively.

### 2.7. Effect of TS40 on Target Bacterial Morphology

We observed that TS40 was able to inhibit bacterial growth. Therefore, to further investigate its effect on cell structure, the morphological changes of *L. anguillarum* after treatment with TS40 were observed by Transmission electron microscopy (TEM). *L. anguillarum* could be observed with a long monopolar flagella and smooth cell surface under a normal physiological state (Figure 5A). In contrast, at 2 h post-TS40-treatment, the flagellum disappeared, and certain cell contents flowed out (Figure 5B). Eventually, at 4 h post-TS40-treatment, the cellular contents of *L. anguillarum* were seriously leaked, and the cellular structure was severely damaged (Figure 5C).

### 2.8. Localization of TS40 in Target Bacteria

To further determine whether TS40 could enter into the cell interior, we treated *L. anguillarum* with FITC-labeled TS40, and we observed the accumulation of extracellular and intracellular fluorescence through a fluorescence microscope. The results indicated that fluorescence could be observed both on the surface and in the cytoplasm of the bacteria, proving that TS40 entered the cell cytoplasm. In contrast, FITC-labeled P86P15 displayed no fluorescence accumulation (Figure 6).

### 2.9. Effects of TS40 on Bacterial Genomic DNA 

Based on the result that TS40 can enter the cytoplasm of target bacteria, we further tested the effect of TS40 on genomic DNA. The results showed that TS40 above 10 μM could inhibit DNA migration, and the DNA bands disappeared when the concentration of TS40 was higher than 20 μM (Figure 7A). To further determine the role of TS40 on the disappeared genomic DNA, the reaction mixture was treated at high temperature with Protease K, and the results showed that the band appeared again with partial degradation (Figure 7B), which further proved that TS40 could bind and degrade part of genomic DNA. However, P86P15 had no effect on genomic DNA.

### 2.10. Effect of TS40 on Bacterial Total RNA

Based on the above experimental results, we further investigated whether TS40 had any effect on total RNA. Similar to the results for genomic DNA, when the concentration of TS40 was higher than 5 μM, the phenomenon of gel retardation appeared, and the total RNA band disappeared completely when the concentration of TS40 was above 20 μM. Conversely, there was no change in total RNA in the presence of P86P15 (Figure 8).

### 2.11. Effect of TS40 on Pathogens Infection 

We analyzed bacterial loads in tissues of fish infected with *L. anguillarum* at different time points. The results showed that the number of *L. anguillarum* in the liver, spleen and kidney of fish injected with TS40 in advance was significantly lower than that of the control group at 24 h (Figure 9). 

Similarly, we measured the copies of RBIV-C1 in the spleen of fish treated with TS40 and found that the viral loads of TS40 group were significantly lower than those of the control group at 5 d post-infection (Figure 10).

### 2.12. Effect of TS40 on Macrophages

In this study, the immunomodulatory potential of TS40 was investigated by examining its effects on macrophage respiratory burst, reactive oxygen species (ROS) production and the expression of immune-related genes. The results showed that TS40 at 400 μM and 600 μM could significantly enhance the respiratory burst of macrophages (Figure 11A), while 600 μM of TS40 could also significantly promote the production of ROS (Figure 11B). The immune genes expression results displayed that the expression levels of heat shock protein 70 (*HSP70*) and serum amyloid A (*SAA*) were significantly increased; the expression levels of tissue necrosis factor (*TNF*) 13B were slightly increased (Figure 11C).

### 2.13. Effect of TS40 on the Proliferation of Peripheral Blood Leukocytes

To detect the effect of TS40 on peripheral blood leukocytes, TS40 was incubated with peripheral blood leukocytes; the MTT assay indicated that 80 μM of TS40 incubation could improve the proliferation of peripheral blood leukocytes significantly (Figure 12). 

## 3. Discussion

In this study, we analyzed and characterized a TFPI-2 homologue, *SsTFPI-2*, from black rockfish, and we investigated the antimicrobial activity, action mode and immune regulation characteristics of TS40, the C-terminal derived peptide of SsTFPI-2. Structurally, there are three KDs in SsTFPI-2, which connected successively between the N-terminal and C-terminal, and these are the typical structural features of TFPI-2 homologues, from teleost to human [22,24]. Moreover, it was inferred that KD1 may be the main functional domain of TFPI-2 [25]. SsTFPI-2 shares 80.5~97.8% amino acid sequence identities with teleost TFPI-2. A phylogenetic analysis showed that *SsTFPI-2* was closest to *S. umbrosus* TFPI-2. With the high sequence identity as well as the phylogenetic analysis and structural features, *SsTFPI-2* was determined to be a new member of the vertebrate TFPI-2 subfamily.

We examined the expression profiles of *SsTFPI-2* in various tissues of black rockfish under normal physiological conditions, and the result showed that *SsTFPI-2* expression was distributed in all the examined tissues, with it being highly expressed in the liver and spleen. The similar tissue distribution results were reported in red drum [26], tongue sole [13] and flounder [22]. The highest expression of TFPI-2 was also found in the liver in both red drum and flounder, which may be related to the role of TFPI-2 in tissue factor pathway-mediated coagulation [27]. Increasing studies have shown that teleosts TFPI-2 participated in antimicrobial immunity. Several studies had reported that bacterial or viral infection could induce *TFPI-2* expression in different tissues [13,22,26]. Similarly, we also found that the expression of *SsTFPI-2* was significantly induced by infection with *L. anguillarum*. Considering the antimicrobial activity of TFPI-2 from humans and teleosts [28,29], the induction of TFPI-2 expression implied the possible participation of this molecular inhibitor in the host innate immunity against microbial evasion.

At present, a series of studies have shown that TFPI C terminal-derived peptides from fish and humans possess antimicrobial and antifungal effects [28,30,31]. EDC34, a peptide derived from human TFPI-1, was active against both Gram-negative and Gram-positive bacteria, as well as pathogenic fungi [32]. Our previous studies have shown that TC24 [33], TC38 [12], TO17 [34], TO24 [21], TP25, TP26 [22] and TC26 [35], which were peptides derived from the C-terminal sequences of tongue sole TFPI-1, tongue sole TFPI-2, red drum TFPI-1, red drum TFPI-2, flounder TFPI-1, flounder TFPI-2 and common carp TFPI-1, displayed a broad spectrum of antibacterial activities, and some of them were also active against megalocytivirus. In the present study, we found that TS40 was antibacterial against both Gram-positive bacteria *S. aureus* and *S. agalactiae* and Gram-negative bacteria *L. anguillarum* and *V. parahaemolyticus*; among them, *L. anguillarum* and *V. parahaemolyticus* are the main pathogens of marine fish. In addition, similar to TO24, TC26 and TC24 [21,33,35], it was found that TS40 had a stronger antibacterial effect against Gram-positive bacteria than Gram-negative bacteria, which may be due to the different composition of the cell membrane [36,37]. These results indicate that TS40 is a new and effective peptide with a broad antibacterial spectrum.

The bactericidal mechanism of most AMPs first acts on the bacterial cell membrane. Baindara found [38] that Lateralsporin 10, a bacteriocin-type antimicrobial peptide, was a membrane-permeable polypeptide, which could disintegrate the cell membrane and subsequent completely lysed the Mtb H37Rv strain. Gu found that BO18 [39], a peptide derived from the bactericidal permeability-increasing protein (BPI) of rock bream, could damage the cell membrane structure of *Vibrio alginolyticus*, cause the content to flow out and, finally, completely destroy the cell structure. Likewise, we observed by TEM that the treatment of TS40 made *L. anguillarum* show similar changes in cells structure, that is, the cell morphology of *V. anguillarum* ranged from the uncomplete cell membrane and the leakage of cellular contents to the completely collapsed whole cell structure. Consistently, our previous reported TFPI-derived peptides, such as TO24 [21], TC24 [33], TO17 [34] and TC26 [35], also destroyed the cell membrane integrity of the target bacteria in a time-or concentration-dependent manner. However, TC38 treatment caused obvious leakage of the cellular contents of *Vibrio vulnificus*, without obvious damage to the cell membrane [12]. These results demonstrated that the interaction between TS40 and the cell membrane of *L. anguillarum* is similar to that of most known TFPI peptides.

It has been found that the interaction between AMPs and nucleic acids is one of the mechanisms by which AMPs play an antibacterial role after entering the target bacteria. In previous reports, we found that TC38 [12], TO24 [21], TP26, TP25 [22], TC26 [35] and TC24 [33] could penetrate the cytoplasm and bind to genomic DNA and total RNA. Similarly, with these features, in this study, we found that TS40 could enter the interior of target bacteria. Then, the interaction between TS40 and genomic DNA showed that TS40 could bind and degrade the genomic DNA of target bacteria in vitro, and the action mode of TS40 on the total RNA was possibly the same as that of genomic DNA. Simultaneously, we speculated that the disappearance of genomic DNA bands in the gel electrophoresis (Figure 8A) might be caused by the absence of a binding site for the nucleic acid dye after the complete binding of TS40 to nucleic acids; after TS40 in the mixture was degraded by proteinase K, the genomic DNA band appeared again with partial degradation (Figure 8B); thus, we predicted that TS40 might completely degrade the genomic DNA when the incubation time and the concentration of TS40 is enough. 

As well as their direct antimicrobial effects, some AMPs can reduce infections by increasing host defense against microbes. Thus, the use of AMPs in mice or mouse models could decrease infection of *E. coli* (CFT073) or *S. aureu* rates in mammals [40,41,42,43]. Oral administration or injection of Epinecidin-1 could significantly enhance the survival rate of zebrafish and grouper infected with *V. vulnificus* [44]. Similarly, in previous studies, we found that TC38, TO24, TC26, TO17 and TC24 administration could reduce the infection of pathogenic bacteria and ISKNV in vivo [12,21,32,33,35]. Consistently, in the present study, and the in vivo analysis showed that the viral copies of RBIV-C1 and the bacterial loads of *L. anguillarum* were significantly lower in TS40-administered fish than in control fish, indicating that TS40 also plays an active role in host defense against pathogenic invasion.

Previous studies have confirmed that AMPs have immunomodulatory effects on the host. In humans, LL-37 could not only enhance the respiratory burst capacity of macrophages [6,45], but also induce the generation of ROS in macrophages [46]. In fish, BO18 enhanced the respiratory burst ability of macrophages and increased expression of immune-related genes in vitro [39], and both dicentracin-like AMP from *Asian sea bass* and moronecidine-like AMP from *Hippocampus* induced higher levels of ROS [47]. Similarly, we found that the respiratory burst ability and ROS of macrophages was significantly enhanced in a dose-dependent manner after TS40 stimulation in vitro. Meanwhile, we found that TS40 could significantly increase the expression of *TNF13-B*, *SAA* and *HSP70*. Among them, *TNF13B* is known to induce B-cell survival and proliferation, immunoglobulin secretion and even enhance immune responses [48]; *SAA*, a positive acute phase protein, could effectively function in signal transduction and eliminate invasive pathogens [49], and *HSP70* is involved in acquired thermal and oxidative tolerance [50]. Furthermore, in this study, leukocyte proliferation could be promoted by an appropriate concentration of TS40. Studies have shown that peptides from swine blood could enhance the respiratory burst capacity of macrophages and effectively improve the proliferation of leukocytes [51]. These above results together confirmed the immunomodulatory function of TS40, which might at least partially account for its in vivo ability to resist pathogen infection.

## 4. Materials and Methods

### 4.1. Experimental Animal and Sample Collection

Healthy black rockfish (*S. schegeli*, average 11.8 ± 1.4 g) and turbot (*Scophthalmus maximus*, average 25.8 ± 1.6 g) were purchased from a commercial farm in the Shandong Province, China, and kept at 19 °C in aerated seawater, which was changed daily. Fish were domesticated for two weeks and feed once a day, before experiment operation. Prior to the experimental manipulation, the presence of pathogens in tissues was verified by the random sampling of fish samples using the previously reported methods [52,53]. The fish were euthanized with excess tricaine methanesulfonic acid (Sigma, St. Louis, MO, USA), prior to tissue collection [30].

### 4.2. Bacteria

*Pseudomonas putida*, *Serratia marcescens*, *V. alginolyticus*, *Vibrio ichthyoenteri*, *Bacillus subtilis*, *Vibrio harveyi*, *Vibrio**litoralis*, *Vibrio parahaemolyticus*, *S**. agalactiae* and *L. anguillarum* were all laboratory preserved strains. The Marine Culture Collection of China (Xiamen, China) provided the strain of *V**. vulnificus*. Tiangen (Beijing, China) provided the strain of *E. coli*. China General Microbiological Culture Collection Center (Beijing, China) provided strains of Micrococcus luteus and *S**. aureus*. *E. coli*, *P. putida*, *S. agalactiae*, *B. subtilis* and *S. aureus* were cultured in Luria-Bertani (LB) medium at 37 °C, and other strains were cultured at 28 °C in LB medium [54]. Megalocytivirus RBIV-C1 (rock bream iridovirus, the first isolated strain in China) was kindly provided by the Institute of Oceanology, Chinese Academy of Sciences [53]. 

### 4.3. Expression of SsTFPI-2 in Fish Tissues under Normal Physiological Conditions 

Total RNA was extracted from five black rockfish tissues using RNAprep Pure Tissue Kit (Tiangen, Beijing, China). Then, the FastQuant RT Kit (With gDNase, Tiangen, Beijing, China) was used for reverse transcription to obtain cDNA. As previously described [35], qRT-PCR was performed in a LightCycler 96 system (Roche Applied Science, Indianapolis, Indianapolis, IN, USA) using the SYBR Green Premix Pro Taq HS qPCR Kit (AG, Changsha, Hunan, China). The primers were listed in Table 3, in which elongation factor 1α (*EF1α*) of black rockfish was used as a reference gene. The expressions of *SsTFPI-2* were analyzed by the 2^−ΔΔCT^ method [55]. The tissue in which the *SsTFPI-2* level was the lowest was set as the control.

### 4.4. Expression of SsTFPI-2 upon Bacterial Infection

As previously reported [56], the black rockfish was infected with *L. anguillarum*. To put it simply, *L. anguillarum* was cultured to OD (Optical density) _600_ = 0.8, washed with PBS and suspended to 1 × 10^6^ CFU/mL. Black rockfish were randomly divided into 3 groups, with 30 fish in each group, and administrated, by intraperitoneal injection of (i.p.), 100 µL *L. anguillarum* or PBS (as control) per fish. Under aseptic conditions, the liver, spleen and head kidney of 5 fish were collected at 0 h, 4 h, 8 h, 12 h, 24 h, 48 h and 72 h after infection. The detection of the expression level was carried out as described above. Primer sequences are shown in Table 3.

### 4.5. Bioinformatics Analysis

The ORF sequence of the *SsTFPI-2* gene was obtained through the previous sequencing of the black rockfish genome in our laboratory [57]. The physicochemical properties of SsTFPI-2 were analyzed by ProtParam. The SMART program was used to determine the region of the signal peptide and protein domains. The secondary structure was analyzed by the Pôle Bioinformatique Lyonnais (PBIL) server. The three-dimensional structure was predicted by protein homology/similarity recognition engine V2.0 (Phyre2, http://www.sbg.bio.ic.ac.uk/phyre2/html/page.cgi?id=index, (accessed on 2 November 2021)). In order to construct the multiple alignments and phylogenetic tree, ClustalX (Higgins D.G., Heidelberg, Germany) and MEGA 5.0 softwares(Sudhir Kumar, Tucson, AZ, USA) were used.

### 4.6. Peptides

The amino acid sequence of the C-terminus of SsTFPI-2 was predicted according to the characteristics, including charge number, isoelectric point and hydropathicity, etc., to obtain an active derived peptide. Then, 5’-FITC-labeled and unlabeled TS40, as well as a negative control peptide P86P15 (FKFLDNMAKVAPTEC), which was derived from the 22–36 residue of P86 of RBIV-C1 [56], were synthesized by China Peptides (Suzhou, China). The peptides were separated and purified by high-performance chromatography with a purity of more than 95%. Before using, these peptides were dissolved in phosphate-buffered saline (PBS, pH 7.4) and stored at a temperature of −80 °C. 

### 4.7. Antibacterial Spectrum

The determination of the antibacterial spectrum was performed according to a previously described method [57]. The above bacteria were cultured in LB to an OD_600_ of 0.8. A total of 50 μL bacteria solution was coated evenly on the LB plates; then, 5 μL of TS40 or P86P15 was dripped into the center of the sterile filter papers placed on the plates. After 24 h of culturing, antibacterial activity was evaluated using the inhibition zone technique.

### 4.8. Antibacterial Activity Assay

The MIC and MBC assay of TS40 were performed as reported previously [35]. Briefly, the bacteria were washed and resuspended in LB to 2 × 10^5^ CFU (colony forming unit)/mL. A series of peptide solutions were obtained by the double dilution method. In 96-well plates, 50 μL of serially diluted peptides and 50 μL of bacterial suspension were mixed and incubated for 24 h at a suitable temperature (as described above). P86P15 or PBS were used as the control group. MIC was defined as the minimum peptide concentration that prevented growth, and the peptide concentration without any bacteria colony present by colony counting was determined as the MBC.

### 4.9. Killing Kinetics Assay

The killing kinetics assay was conducted using methods reported previously, with a little modification [35]. In brief, *L. anguillarum* and *S. aureus* were cultured as described above. Bacteria were washed with LB and suspended to 2 × 10^5^ CFU/mL. The target bacteria were incubated with 5 × MIC TS40, P86P15 or PBS. Samples were taken every hour, diluted appropriately in PBS and coated on a plate, and this was repeated three times. After incubation for 24 h, the number of bacterial colonies on the plates was counted.

### 4.10. TEM Assay

This experiment was based on the method previously reported [58]. The washed *L. anguillarum* were resuspended to 1 × 10^10^ CFU/mL by PBS. A total of 50 μL of bacterial cells were exposed to 4 × MIC TS40 at 28 °C for 0 h, 2 h and 4 h, respectively. After that, the cells were fixed with 2.5% glutaraldehyde and deposited on carbon-coated copper girds. Then, the grids were dried naturally and negatively stained with phosphotungstic acid. Finally, the grids were observed with a transmission electron microscope (TEM, GEM-1200, GEOL, Akishima-shi, Tokyo Metropolis, Japan).

### 4.11. Fluorescence Microscopy

This experiment was carried out as previously reported [12]. *L. anguillarum* was cultured and resuspended to 1 × 10^7^ CFU/m. FITC-labeled TS40 or P86P15 were reacted with the bacterial cells at 28 °C for 1 h. After washing with PBS, a fluorescence microscope (Leica DM 2500, Witzlar, Hesse, Germany) was used to observe the extracellular fluorescence. A total of 0.4% Trypan blue was added to the mixture and incubated for 30 min to quench extracellular fluorescence, and intracellular fluorescence was observed as above.

### 4.12. Effect of TS40 on Genomic DNA

This assay was carried out using previously reported methods [34,57]. Briefly, the TIANamp Bacteria DNA Kit (Tiangen, Beijing, China) was used to extract DNA from *L. anguillarum*. A total of 100 ng of DNA was mixed with 2 μL of TS40 or P86P15, and DNA electrophoresis was performed after the mixture was incubated in a 25 °C water bath for 30 min. For further analysis of the disappearing genomic DNA band in the mixture, 2 μL Proteinase K was added and incubated at 70 °C for 15 min before DNA electrophoresis.

### 4.13. Effect of TS40 on Total RNA

The effect of TS40 on total RNA of bacteria was studied using previously reported methods [33]. Total RNA of *L. anguillarum* was extracted by the RNAprep pure Bacteria Kit (Tingke, Beijing, China). A total of 100 ng of total RNA of *L. anguillarum* was mixed with TS40 or P86P15, and RNA electrophoresis was performed after incubating in a 25 °C water bath for 30 min. 

### 4.14. In Vivo Study on Pathogens Infection

The effects of TS40 on bacterial invasion in black rockfish was investigated as previously reported [13]. Briefly, black rockfish were divided into 3 groups, with 20 fish in each group. The TS40 group were administrated 600 μM TS40 at a dose of 100 μL via i.p. injection. The control groups were administrated an equal volume of 600 μM P86P15 or PBS, respectively. After 1 h, the fish were infected i.p. with *L. anguillarum* (2 × 10^5^ CFU/fish). At 12 h and 24 h post-infection, the liver, spleen and kidney were taken under aseptic conditions and then weighed, homogenized and diluted with PBS, before plating in triplicate on LB agar plates. The plates were incubated at 28 °C for 24 h, and the colonies were counted and calculated as the number of per milligram. Three parallel data were set in the experiment, and the data were averaged.

According to a previous report [13], the effect of TS40 on virus invasion was studied. Because turbot is susceptible to RBIV-C1, while black rockfish was not susceptible, turbot was used and grouped as described above. Before infection, the tissue homogenates containing RBIV-C1 were diluted 10 times and then mixed with 500 μM TS40 and incubated at 22 °C for 6 h. According to research, the spleen was the mainly target organ infected by RBIV-C1, so spleens of the fish were taken under aseptic conditions at 1 d, 3 d and 5 d post-infection. RBIV-C1 copy numbers in the spleen were determined by absolute quantitative real time PCR as reported previously [52]. The PCR primers are shown in Table 3.

### 4.15. Effect of TS40 on Macrophages

#### 4.15.1. Determination of Respiratory Burst

Macrophages were isolated by the Fish Tumor Tissue Macrophage Isolation Kit (TBD, Tianjin, China) from the head kidney and spleen of black rockfish. The isolated macrophages were resuspended to 1 × 10^8^ cell/mL with a 1640/dual anti-cell culture medium. A total of 100 μL of macrophages were added in each well of 96-well polypropylene microtiter plates and cultured at 25 °C for 24 h. Then, the respiratory burst of macrophages was measured as previously reported [56]. Briefly, the cells were washed three times with PBS, and then, they were mixed with TS40 at final concentrations of 0 μM, 20 μM, 80 μM, 200 μM, 400 μM and 600 μM, respectively. The cells were washed with PBS for three times again after incubation at 25 °C for 3 h. The respiratory burst of macrophages was determined by a previously reported method [59]. The plates were shaken on an enzyme standardizer for 5 min; after that, the absorbance value was determined at 630 nm, and the results were analyzed.

#### 4.15.2. Detection of Reactive Oxygen Species (ROS)

The production of ROS within macrophages was measured by a 2′,7′-dichlorofluorescin-diacetate (DCFH-DA) assay, as described, with slight modification [46]. Briefly, the above isolated macrophages were resuspended and incubated with different concentrations of TS40 (0 μM, 20 μM, 200 μM and 600 μM) for 1.5 h at 25 °C; then, the cells were treated with DCFH-DA; after incubation for 20 min, the cells were washed twice with a serum-free medium. Finally, the fluorescence value was detected by a multi-function enzyme-labeled instrument (SpectraMax iD5, San Jose, CA, USA) at OD_488_ and OD_525_.

#### 4.15.3. Expression Analysis of Immune-Related Genes

This analysis was carried out according to the previously report [39]. Briefly, the isolated macrophages were laid on a 24-well plate; then, 600 μM of TS40 was added into the wells and incubated with the cells for 2 h. After that, the cells were collected, and the total RNA extraction and cDNA synthesis were conducted as above. To determine the expression of immune genes, including interleukin (*IL*)-*1β*, tissue necrosis factor (*TNF*) *13B*, haptoglobin (*HP*), serum amyloid A (*SAA*), heat shock protein 70 (*HSP70*) and *ISG15* (an interferon-stimulated gene), a qRT-PCR was performed as above. PCR primers of the immune genes reported previously [60] are shown in Table 3. 

### 4.16. Effect of TS40 on Peripheral Blood Leukocytes Proliferation

For the in vitro study, black rockfish caudal vein blood samples were treated with heparin sodium under sterile conditions, and the peripheral blood leukocytes were isolated by a previously reported method with slight modifications [61]. After being cultured at 37 °C, the cells were stimulated overnight with 50 μL of 20 μM, 80 μM, 200 μM and 400 μM of TS40. P86P15 was used as the control group, and the mixture was washed with PBS for three times. The proliferation of peripheral blood leukocytes was detected by the MTT method [62]. Finally, the OD_540_ was determined and analyzed.

### 4.17. Statistical Analysis

All experiments were repeated three times, and data are presented as the means ± SEM. SPSS 25.0 software was used for the analysis of variance (ANOVA) and Duncan’s multiple comparisons. The level of significance was defined as *p* < 0.05.

## 5. Conclusions

We investigated, for the first time, the molecular feature and expression patterns of SsTFPI-2, as well as the biological functions of the C-terminal derived peptide TS40. The results indicated that SsTFPI-2 was expressed ubiquitously in multiple tissues. After stimulating with *L. anguillarum*, SsTFPI-2 exhibited significant increased expressions. TS40 was active against both Gram-negative and Gram-positive bacteria, and it could destroy the cell membrane, enter into the cytoplasm and interact with nucleic acids. The in vivo analysis indicated that TS40 effectively inhibited infection in fish caused by *L. anguillarum* and RBIV-C1. Furthermore, TS40 increased the respiratory burst, ROS production, and the expression of immune-related genes in macrophages. These results imply the involvement of SsTFPI-2 in fish innate immunity and add new insights into the role of teleost TFPIs. Meanwhile, TS40 possesses application potential in the prevention and control of aquatic animal diseases.

## Figures and Tables

**Figure 1 marinedrugs-20-00353-f001:**
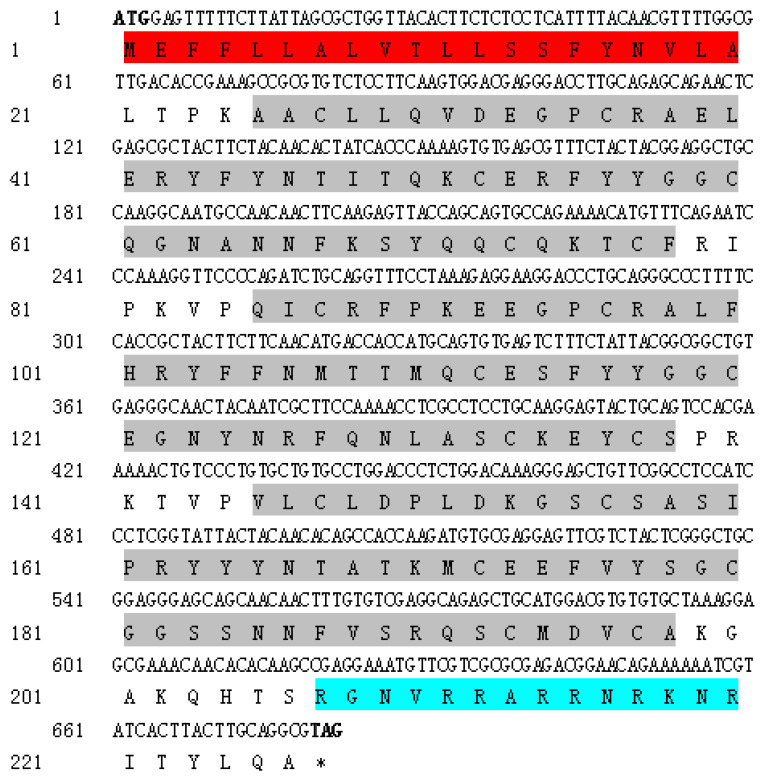
The nucleotide and amino acid sequences of SsTFPI-2. In the cDNA sequence, the translation start and stop codons are in bold. In the amino acid sequence, the signal peptide is shaded in red, the three Kunitz domains are shown in shades of grey and the low complexity sequence is shaded blue. * Represents the stop codon.

**Figure 2 marinedrugs-20-00353-f002:**
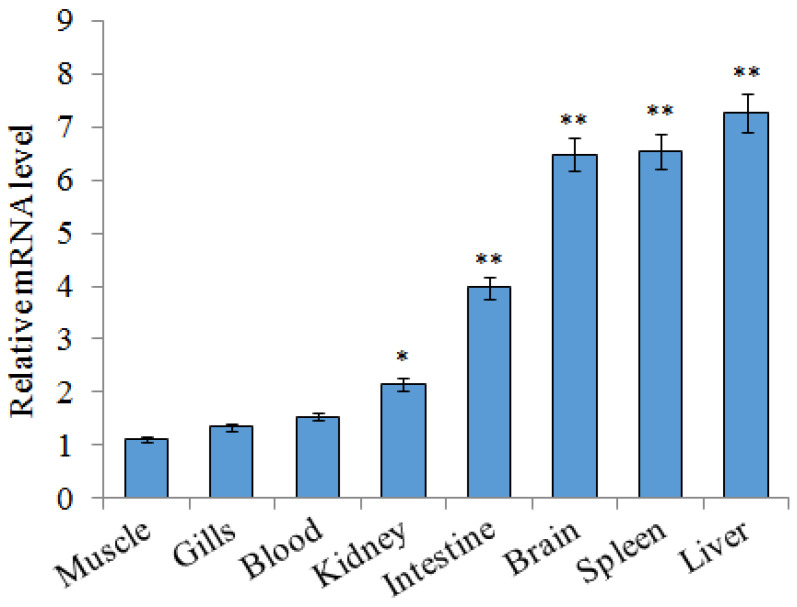
*SsTFPI-2* expression in black rockfish tissues. *SsTFPI-2* expressions in various tissues were determined by qRT-PCR. Each result is the average of three independent experiments, shown as means ± SEM. * 0.01< *p* < 0.05, ** *p* < 0.01.

**Figure 3 marinedrugs-20-00353-f003:**
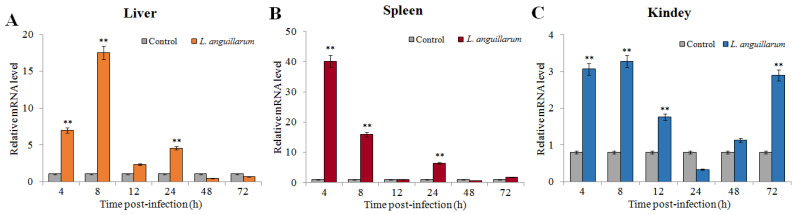
*SsTFPI-2* expression in response to *Listonella anguillarum* challenge. *L. anguillarum* was used to infect black rokerfish, and PBS was used as the control. After infection, the liver (**A**), spleen (**B**) and head kidney (**C**) were taken for aseptic treatment at 4 h, 8 h, 12 h, 24 h, 48 h and 72 h. The *SsTFPI-2* expression in each tissue was determined by qRT-PCR at various time points. Each result is the average of three independent experiments, shown as means ± SEM. ** *p* < 0.01.

**Figure 4 marinedrugs-20-00353-f004:**
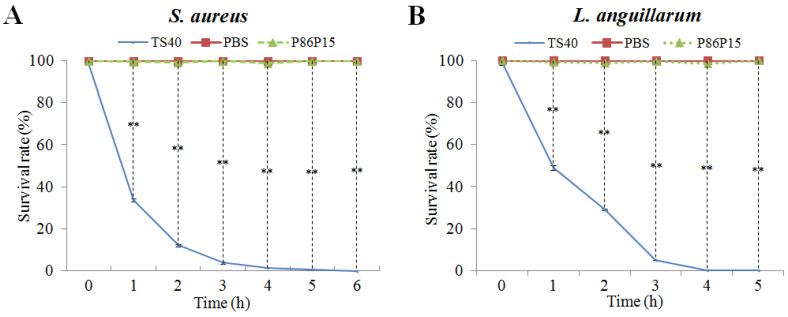
Bacterial killing kinetics of TS40 against (**A**) *Staphylococcus aureus* and (**B**) *Listonella anguillarum*. Each result is the average of three independent experiments, shown as means ± SEM. ** *p* < 0.01.

**Figure 5 marinedrugs-20-00353-f005:**
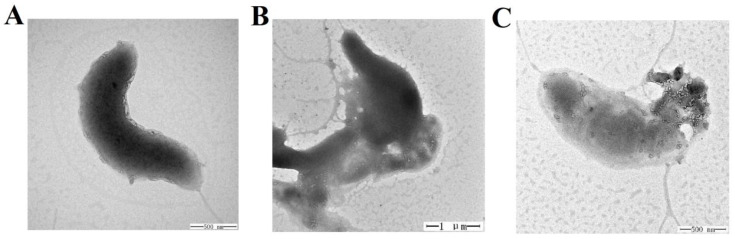
Effect of TS40 on the cell structure of *Listonella anguillarum*. *L. anguillarum* was treated with TS40 for 0 h (**A**), 2 h (**B**) or 4 h (**C**) and then observed by transmission electron microscopy (TEM). Magnification: 60,000× (**A**), 20,000× (**B**), 50,000× (**C**). Scale bar: 500 nm (**A**), 1 µm (**B**), 500 nm (**C**).

**Figure 6 marinedrugs-20-00353-f006:**
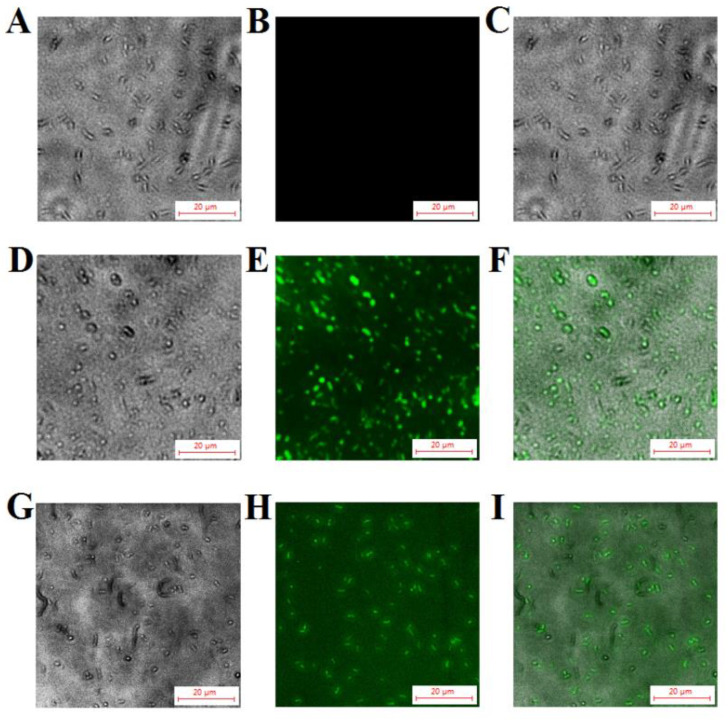
Fluorescent localization of TS40 in target bacteria. *Listonella anguillarum* interacted with FITC-labeled TS40 (**D**,**E**,**G**,**H**) or FITC-labeled P86P15 (**A**,**B**) for 2 h under the fluorescence channel (**B**,**E**,**H**) and white light channel (**A**,**D**,**G**) and was observed after quenching the external fluorescence. (**C**), (**F**), (**I**) are the merged images of (**A** and **B**), (**D** and **E**), (**G** and **H**)**,** respectively. Magnification, 400×. The scale bar is 20 µm.

**Figure 7 marinedrugs-20-00353-f007:**
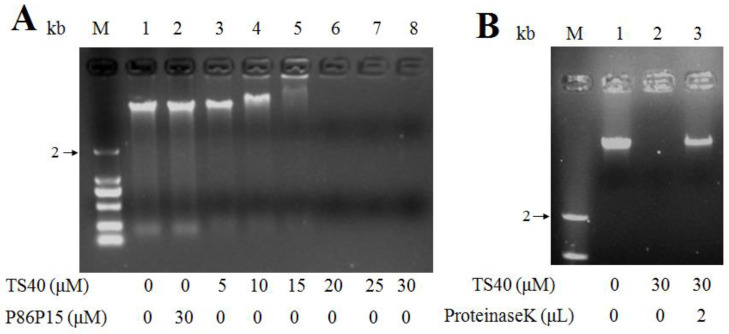
Effect of TS40 on the genomic DNA of *Listonella anguillarum* in vitro. (**A**) Electrophoresis images of *L. anguillarum* genomic DNA incubated with different concentrations of TS40, P86P15 or PBS; (**B**) Electrophoresis images of proteinase K degradation of TS40 at a high temperature.

**Figure 8 marinedrugs-20-00353-f008:**
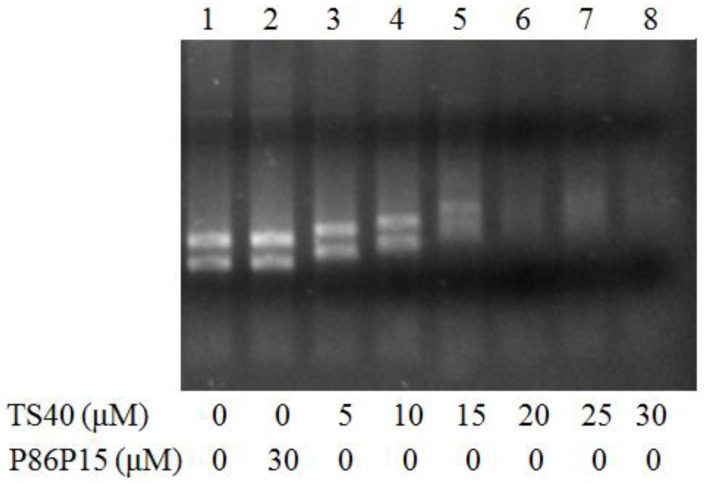
Effect of TS40 on total RNA of *Listonella anguillarum* in vitro. The RNA identification diagram of *L. anguillarum* total RNA treated with TS40, P86P15 or PBS.

**Figure 9 marinedrugs-20-00353-f009:**
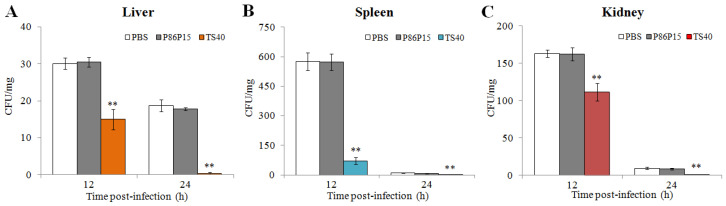
In vivo effect of TS40 on bacterial infection. The bacterial amounts in the liver (**A**), spleen (**B**) and kidney (**C**) of different treatment groups were determined after infection. Each result is the average of three independent experiments, shown as means ± SEM. ** *p* < 0.01.

**Figure 10 marinedrugs-20-00353-f010:**
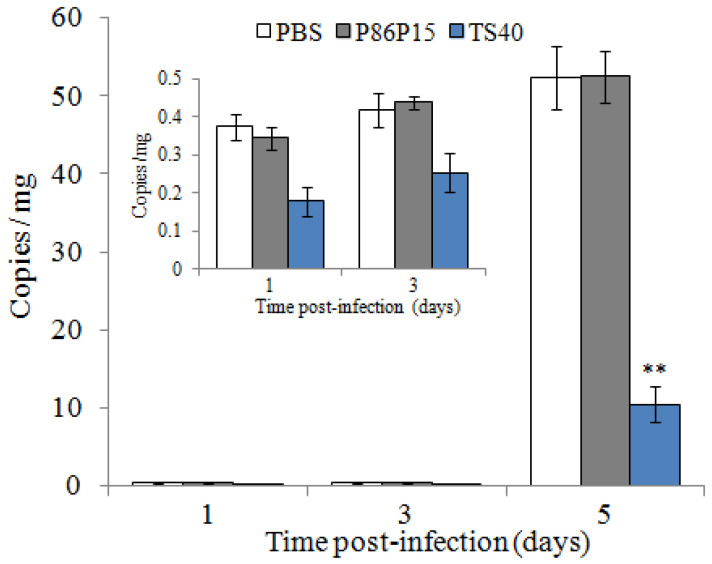
Effect of TS40 on virus infection. Virus copies at each time point in the spleen of fish after treatment with RBIV-C1 under different conditions. Each result is the average of three independent experiments, shown as means ± SEM. ** *p* < 0.01.

**Figure 11 marinedrugs-20-00353-f011:**
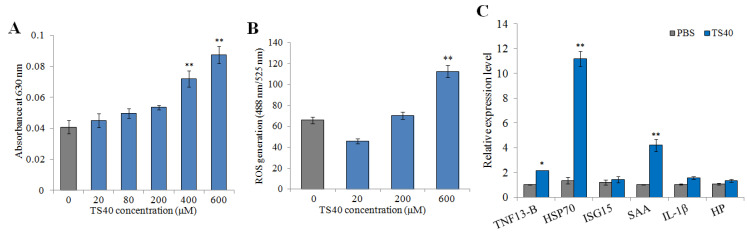
Effect of TS40 on the activity of macrophages. The effects of different concentrations of TS40 on (**A**) respiratory burst, (**B**) reactive oxygen species (ROS) and (**C**) immune-related gene expression. Each result is the average of three independent experiments, shown as means ± SEM. * 0.01 < *p* < 0.05, ** *p* < 0.01.

**Figure 12 marinedrugs-20-00353-f012:**
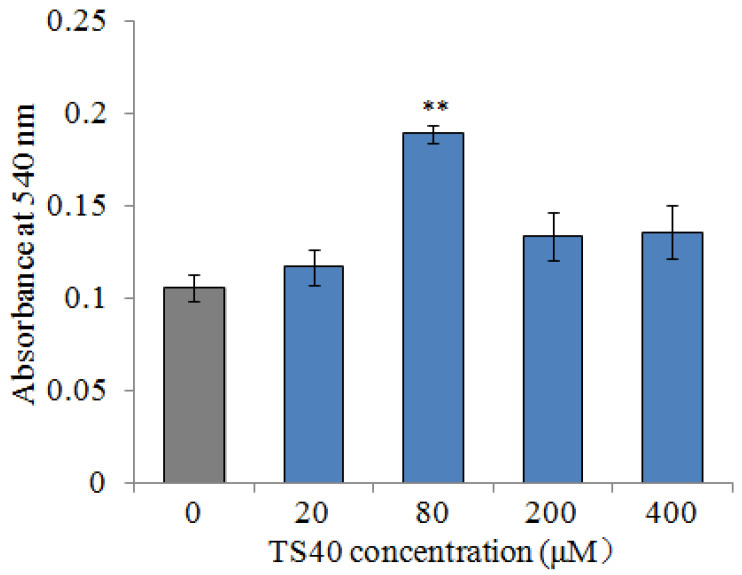
The proliferation effect of TS40 on the peripheral blood leukocytes. The proliferation activity of cells treated with different concentrations of TS40 or P86P15 was measured by an MTT assay. Each result is the average of three independent experiments, shown as means ± SEM. ** *p* < 0.01.

**Table 1 marinedrugs-20-00353-t001:** Physical and chemical properties analysis of SsTFPI-2.

Properties	SsTFPI-2	TS40
Number of amino acids	226	40
Total number of atoms	3571	654
Formula	C_1141_H_1754_N_324_O_329_S_23_	C_192_H_329_N_75_O_55_S_3_
Molecular weight (MW/Da)	26,011.83	4664.37
Theoretical isoelectric point (PI)	9.14	11.79
Grand average of hydropathicity (GRAVY)	−0.462	−1.087
Instability index (II)	43.32	86.90
Aliphatic index (AI)	55.27	51.25
Total number of negatively charged residues (ASP + GLU)	16	1
Total number of positively charged residues (ARG + LYS)	31	11

**Table 2 marinedrugs-20-00353-t002:** Minimum inhibitory concentration (MIC) and minimum bactericidal concentration (MBC) of TS40 against bacteria.

Strains	MIC (μM)	MBC (μM)
*Listonella anguillarum*	25	50
*Vibrio parahaemolyticus*	400	>800
*Staphylococcus aureus*	12.5	25
*Streptococcus agalactiae*	800	>800

**Table 3 marinedrugs-20-00353-t003:** Primers used in the experiment.

Primers	Sequences (5′–3′)
SsTFPIRTF	TCCCAAAGGTTCCCCAGAT
SsTFPIRTR	CTCACAGCCGCCGTAATAGA
MCPRTF	CATCAGCCAGAGCACCCAG
MCPRTR	ACCTCACGCTCCTCACTTGTC
EF1α-F	AACCTGACCACTGAGGTGAAGTCTG
EF1α-R	TCCTTGACGGACACGTTCTTGATGTT
TNF13B-F	GGAAAACCTTCAGGAAAGAATACA
TNF13B-R	TGAGGCTCGTCTCCCACC
IL-1β-F	GCATCCGAGGCACAAATCC
IL-1β-R	ACACCCGCTCCACTCAACAG
HP-F	GGCAGGGAAAGAGGGAATAG
HP-R	GGAAGTGTGGATGGAGAAAAA
SAA-F	CTTCCCCGGTGAAGCCTTTA
SAA-R	CCATGCTCATTTGCTCTCTGAT
HSP70-F	CTGTTTGAAGCAATTGAGGGC
HSP70-R	CAGGAGTTTCTGGATTTTAGGGA
ISG15-F	CTACGGCCTGCAGCAAGGAGC
ISG15-R	CCCTGGTCTTGAAGTTGGCCA

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
