# Peer review of "Antimicrobial and Immunoregulatory Activities of TS40, a Derived Peptide of a TFPI-2 Homologue from Black Rockfish (Sebastes schlegelii)"

_marinedrugs, 2022, doi:10.3390/md20060353_

Round 1

Reviewer 1 Report

  1. General comments

In the manuscript, the molecular feature and expression patterns of SsTFPI-2, as well as the biological functions of the C-terminal derived peptide TS40 were investigated, and TS40 showed a bactericidal effect on Gram-negative and Gram-positive bacteria. Furthermore, TS40 showed its antibacterial activity by enhancing the cell membrane permeability, destroying cell structure and entering the cytoplasm to interact with nucleic acids. The result suggests that TS40 has the potential to be served as a novel therapeutic agent in aquaculture.

  1. Major revision

As EDC34 was released in human wounds, and can be generated by neutrophil elastase from human TFPI-2 in ref.18 of the manuscript, the reader can easily understand why the author focuses on EDC34. It is strongly recommended to explain the reason why the author focuses on the amino acid sequence of 187-226 in SsTFPI-2, TS40, including the physical and chemical properties of TS40, in addition to its secondary structure.

  1. Minor revision

Line 148: Revise “membrane permeabilization ability of TS40 wase estimated” to “membrane permeabilization ability of TS40 was estimated”.

Line 188: Revise “subjected to electrophoretic” to “subjected to electrophoretic analysis”.

Reviewer 2 Report

  1. What is TS-40 and what is SsTFPI-2? Are they different? What is the peptide sequence shown in the abstract? Please write it clearly.
  2. What was the control for the gene expression experiment?
  3. How do authors determine that the effect of TS-40 with rifampicin and erythromycin was synergistic? Please justify? Here is one reference. https://pubmed.ncbi.nlm.nih.gov/27267959/
  4. Line139, how this assay determines the outer membrane permeability is not clear? Please explain and justify.
  5. What about the killing mechanism for S. aureus? Authors should do the assays for S. aureus too, same as Listonella anguillarum.
  6. Figure6, why authors chose 2 hours of treatment time when only 50% survival is observed at a 1-hour time point? It is recommended to do microscopy experiments at lower time points and lower MIC values to see the clear effects on the bacterial membrane.

Round 2

Reviewer 2 Report

1.       Please include the statistics in Fig2 and 4, as well in their legends too.

2.       Fig5. Please mention TEM or SEM.
